# Rapid and Local Self-Healing Ability of Polyurethane Nanocomposites Using Photothermal Polydopamine-Coated Graphene Oxide Triggered by Near-Infrared Laser

**DOI:** 10.3390/polym13081274

**Published:** 2021-04-14

**Authors:** Yu-Mi Ha, Young Nam Kim, Yong Chae Jung

**Affiliations:** Institute of Advanced Composite Materials, Korea Institute of Science and Technology (KIST), 92 Chudong-ro, Bongdong-eup, Wanju-gun, Jeollabuk-Do 55324, Korea; yumiha87@gmail.com (Y.-M.H.); t15930@kist.re.kr (Y.N.K.)

**Keywords:** self-healing materials, polyurethane, near-infrared light, photothermal effect, disulfide bonds, polydopamine, reduced graphene oxide

## Abstract

In this study, we report the self-healing ability of polyurethane (PU) nanocomposites based on the photothermal effect of polydopamine-coated graphene oxide (PDA–rGO). Polydopamine (PDA) was coated on the graphene oxide (GO) surface, while simultaneously reducing GO by the oxidation of dopamine hydrochloride in an alkaline aqueous solution. The PDA–rGO was characterized by Fourier-transform infrared spectroscopy, X-ray diffraction, Raman spectroscopy, thermogravimetric analysis, and scanning electron microscopy–energy-dispersive X-ray analysis. PDA–rGO/PU nanocomposites with nanofiller contents of 0.1, 0.5 and 1 wt% were prepared by ex situ mixing method. The photothermal effect of the PDA–rGO in the PU matrix was investigated at 0.1 W/cm^2^ using an 808 nm near-infrared (NIR) laser. The photothermal properties of the PDA–rGO/PU nanocomposites were superior to those of the GO/PU nanocomposites, owing to an increase in the local surface plasmon resonance effect by coating with PDA. Subsequently, the self-healing efficiency was confirmed by recovering the tensile stress of the damaged nanocomposites using the thermal energy generated by the NIR laser.

## 1. Introduction

Self-healing materials possess the inherent ability to recover from damage, thus maintaining their function and structure. As a result, products composed of smart materials have an extended lifetime, and any additional maintenance costs associated with long-term use are avoided [1]. In general, self-healing is initiated by molecular diffusion, contact of failure surfaces, dynamic bonding, and the formation of interactions [2,3].

To recover the shape and heal the damaged parts, materials are needed as an energy source to facilitate molecular diffusion. Many self-healing methods rely on energy sources, such as heat [4], light [5], and electricity [6]. Most studies have focused on self-healing systems with thermal energy as the fundamental energy source; thermal energy can easily repair damaged polymers [7,8]. However, thermal energy can also affect the undamaged parts of the polymers, which can cause dissociation of the dynamic bond network. Consequently, the material properties deteriorate due to heating, which may negatively impact the quality of the products.

Polyurethane (PU) has excellent flexibility, mechanical properties, high elasticity, and biocompatibility [9]. PU has a typical hard/soft segmental, dual-component structure. PU has shape memory properties, due to the thermodynamic incompatibility between hard and soft segments. The shape memory effect can help the self-healing ability [7]. Therefore, PU has attention as a candidate for self-healing polymers. Self-healing PU based on reversible bond systems, including the Diels–Alder reaction [10], disulfide bonds [7,11], and a hydrogen bond [12] were reported. Hernandez et al. reported the self-healing properties of nanocomposites. Self-healing efficiencies linearly increased according to graphene content, but the correlation was not complete. The healing efficiency of the nanocomposite is due to the interaction of polymer–graphene [13].

Near-infrared (NIR) irradiation is another method of repairing damaged materials [14]. NIR spectrometers enable adjustment of the distance to the sample and the laser power, allowing locally high precision and high efficiency to repair damaged specimens [15,16]. In particular, NIR light is harmless to the human body [17]. As one of the methods to fabricate materials with photothermal effects that respond to NIR lasers, carbon materials, such as graphene and carbon nanotubes (CNTs), are introduced into the polymer matrix as reinforcements [9]. Graphene has been widely used as a matrix reinforcing agent, owing to its excellent thermal, mechanical, and optical properties [18]. In addition, it is known to have excellent photothermal energy conversion capabilities. In particular, graphene exhibits a high capacity (e.g., light absorption section) that converts absorbed light into heat energy by resonating with a NIR laser line [19,20,21].

Recently, a system for polymer composite materials that self-heals via the photothermal effect has been proposed [22,23,24,25]. In the self-healing mechanism induced by NIR, fast heat transfer and strong NIR absorption are important, because the heat generated by the NIR light is rapidly transferred to the center to activate self-healing.

In our previous work [26], we investigated the self-healing ability of polycarbonate-diol-based polyurethane (PU) with disulfide bonds. Disulfide bonds are well-known for undergoing thiol–disulfide dynamic exchange reactions via reversible covalent interactions [7,11]. The PU–disulfide film synthesized according to the soft segment change demonstrated not only excellent mechanical properties but also high self-healing efficiency and shape-recovery properties [27].

In this study, we designed a system for polymer nanocomposites that undergo self-healing via the photothermal effect. First, we synthesized reduced graphene oxide (GO) coated with polydopamine (PDA–rGO) to increase the photothermal effects. PDA–rGO/PU nanocomposites with nanofiller contents of 0.1, 0.5 and 1 wt% were prepared. As the filler content was increased, the mechanical properties and photothermal effects of the nanocomposites were enhanced. Finally, damaged samples of self-healing PU nanocomposites containing PDA–rGO exhibited local and high-efficiency healing by NIR irradiation (Figure 1).

## 2. Materials and Methods

### 2.1. Materials

Polycarbonate diols (M_w_ = 1000 g/mol) and (M_w_ = 3000 g/mol) were purchased from UBE Industries, LTD., Tokyo, Japan. 4,4-Methylene bis(phenyl isocyanate) (MDI), 2-hydroxyethyl disulfide (HEDS), and dopamine hydrochloride were purchased from Sigma–Aldrich. Chloroform and dimethylformamide (DMF) were purchased from Daejung Chemicals & Metals Co., Ltd. Graphene oxide powder (GO-V30, average lateral dimension: ≈30 μm) was purchased from Standard Graphene Inc., Ulsan, Korea. A 1 M tris-HCl solution (pH 8.5) was purchased from Biosesang, Korea.

### 2.2. Synthesis of PDA–rGO

Commercial GO powder (0.3 g) was dispersed in 300 mL of 10 mM tris-HCl solution (pH 8.5) under horn-type ultrasonication for 2 h in an ice bath (at 20 kHz frequency and 150 W power; Sonics). Dopamine hydrochloride (0.15 g) was then added and dispersed by sonication for 15 min in an ice bath. The reaction mixture was stirred for 15 h at room temperature, and the PDA–rGO product was dialyzed using a dialysis membrane (MWCO 6–8 kDa, Spectrum Laboratories, Inc., USA) against deionized water for 72 h. The PDA–rGO powder was obtained by freeze-drying.

### 2.3. Preparation of PDA–rGO/PU Nanocomposites

Self-healing PU was synthesized by condensation polymerization using two different polycarbonate diols as a soft segment and MDI and HEDS as hard segments. For the self-healing PU elastomer, two different polycarbonate diols with M_w_ = 1000 g/mol (22.5 g) and M_w_ = 3000 g/mol (22.5 g) were preheated at 100 °C under vacuum for 24 h to remove moisture before the reaction. Then, MDI (15.02 g) was added along with chloroform and stirred for 3 h. The mixture was cooled to 25 °C, and then HEDS (4.63 g) diluted with DMF was slowly added dropwise to the reaction mixture. The exact molar ratio of the reactants was 1:2:1 (polycarbonate diol:MDI:HEDS). After 2 h, the product was precipitated in 500 mL of cold methanol to remove the unreacted compounds. After washing several times with methanol, it was dried at 30°C in vacuum, and the yield was 95%. The resulting product was a colorless solid (1 g), which was redissolved in chloroform (9 mL). PU nanocomposites with PDA–rGO were prepared by the solvent casting method on glass dishes with 12 cm diameters. PDA–rGO/PU solutions were prepared with PDA–rGO at 0.1, 0.5 and 1 wt%. The calculated amount of PDA–rGO was dispersed in chloroform (20 mL) by horn-type sonication for 30 min, and then the 10 wt% PU solution was added to the PDA–rGO solution. The mixture was stirred for 24 h at room temperature and dried under vacuum at 50 °C for 48 h. The thickness of the resulting film was determined to be approximately 300–400 μm.

### 2.4. Characterization

Raman spectra of GO and PDA–rGO were recorded on an InVia Reflex spectrometer (Renishaw) in the range of 1200–2000 cm^−1^ with a 514 nm laser. The morphologies of GO and PDA–rGO were observed by field-emission scanning electron microscopy (FE–SEM; NOVA NanoSEM 450, FEI). Fourier-transform infrared (FTIR) spectroscopy was performed using a Nicolet iS10 spectrometer (Thermo Scientific) to identify the functionalization of GO. The thermal stability of the PDA–rGO was examined using thermogravimetric analysis (TGA; Q50, TA Instruments, New Castle, DE, USA) in the temperature range of 40–800 °C at a constant heating rate of 10 °C/min under air. The thermal properties of the GO/PU and PDA–rGO/PU nanocomposites were investigated using a dynamic mechanical analyzer (DMA; DMA Q800, TA Instruments) with a heating rate of 3 °C/min at a frequency of 1 Hz and amplitude of 15 µm using rectangular samples with dimensions of 25 (length) × 7 (width) × 0.3 (thickness) mm. The mechanical properties elongation at break, breaking stress, and Young’s modulus were measured at room temperature, according to the ASTM D638 test method, using a universal testing machine (UTM; Instron model 5567A) at a crosshead speed of 20 mm/min in a 100 N load cell. Each sample was measured using a dumbbell specimen with dimensions of 60 (length) × 10 (width) × 20 (narrow part length) × 3 (narrow part width) × 0.3 (thickness) mm. The samples were tested at least four times. The photothermal properties of the nanocomposites were determined by exposing the sample to a NIR laser (PSU-III-LED) line at 808 nm at 0.1 W/cm^2^ for 80 s. The NIR laser was irradiated onto the sample at a distance of 10 cm from the laser source. The heat energy generated in the sample was monitored using an IR camera (FLIR T420, FLIR system). Self-healing tests were performed to determine the damaged sample’s ability to recover its mechanical properties. The film sample was cut to approximately 50% thickness using a razor blade, the damaged surface was healed by NIR laser irradiation, and then the mechanical properties were measured. The self-healing efficiency was calculated by dividing the healed tensile stress of the damaged nanocomposites by the original tensile stress.

## 3. Results and Discussion

### 3.1. Synthesis and Characterization of PDA–rGO

The reaction mechanism of the polydopamine-coated GO synthesized in this study can be schematically illustrated in Figure 2a. Dopamine is spontaneously oxidized and polymerized on the surface of GO in a weakly alkaline, aqueous solution at room temperature, as shown in Figure 2a. PDA–rGO was synthesized via oxidation polymerization using dopamine hydrochloride [28,29]. In other words, this was because the catechol groups in DA were oxidized to quinone groups; these then combined with OH on the surface or edge of GO, resulting in PDA–rGO [30]. This result was confirmed by the FTIR spectroscopy, Raman, and XRD.

Figure 3a shows the FTIR spectra of GO and PDA–rGO. The spectrum of GO exhibits a broad band at 3360 cm^−1^, which is attributed to the stretching vibration of the O–H bond. In addition, the GO spectrum contains several characteristic peaks at 1730, 1610, 1215, and 1045 cm^−1^, which are associated with the stretching vibrations of C=O, C=C, C–O, and C−O−C, respectively [31]. In contrast, the PDA–rGO spectrum showed not only a strong, broad band between 3600 and 2800 cm^−1^ corresponding to –OH but also a characteristic peak at 3230 cm^−1^, corresponding to the N–H bond of PDA. In addition, the characteristic peaks of PDA at 1510 and 1120 cm^−1^ correspond to aromatic N–H and C–N, respectively [32]. Figure 3b shows the X-ray diffraction (XRD) patterns of GO and PDA–rGO. GO exhibited a strong peak at 2θ = 10.3°, indicating that many oxygen-containing groups were intercalated within the interlayer space. This intense GO peak completely disappeared, and a broad peak appeared in the PDA–rGO diffractogram at 2θ = 24.5°, indicating that GO was successfully reduced by PDA [32]. Raman spectroscopy is a powerful tool for detecting crystalline disorders and the degree of chemical modification of carbon nanostructures [33]. The Raman spectra of GO and PDA–rGO were measured using a 514 nm laser line. Both GO and PDA–rGO showed an obvious G band at 1590 cm^−1^ (graphite mode) and D band (defect-induced mode) at 1350 cm^−1^, as shown in Figure 3c. The intensity ratio of the D and G bands (I_D_/I_G_), R value (defined as *I_D_/I_G_*, the integrated intensity of the D band divided by the integrated intensity of the G band), is associated with graphene disorder; therefore, it is used to measure the degree of defects [32]. PDA–rGO showed a slight increase in D band intensity, compared to GO. The R value of GO and PDA–rGO were 0.87 and 0.90, respectively. These results show an increase in edge defects in PDA–rGO due to the grafting of PDA in GO. Finally, the thermal stabilities of GO and PDA–rGO were determined by TGA, as shown in Figure 3d. The measurements were performed at a healing rate of 10 °C/min under air. GO showed a major weight loss of 49 wt% at 160 °C due to the elimination of moisture and oxygen-containing groups, followed by a weight loss of 35 wt% between 400 and 500 °C due to the elimination of carbonyl, hydroxyl, and stable oxygen functionalities [34]. It can be considered that the mass loss above 450 °C is due to the decomposition of the carbon lattice [35].

In contrast, PDA–rGO showed a slow weight loss of 15 wt% up to 240 °C, which represents the elimination of the decomposition of PDA and nonreduced oxygen-containing functional group of GO [36,37]. In addition, we observed rapid weight loss of 58 wt% between 450 and 600 °C due to the elimination of the carboxylic group. These results indicate that PDA–rGO is more thermally stable than GO and confirmed that GO was successfully reduced. In other words, we were able to synthesize polydopamine-coated reduced graphene (PDA–rGO) by reacting GO and dopamine. Furthermore, it can be seen that the pyrolysis behavior of PDA–rGO is very similar to that of typical rGO [38].

The morphologies of GO and PDA–rGO were analyzed by FE–SEM. GO and PDA–rGO powders were less than 100 μm in size. In particular, it was confirmed that the 3.4% N content was uniformly distributed on the PDA–rGO surface through SEM–energy dispersive X-ray analysis (SEM–EDX), as shown in Figure 4 and Appendix A.

### 3.2. Preparation and Characterization of PDA–rGO/PU Nanocomposites

The disulfide bond is reversible by a thiol–disulfide dynamic exchange reaction at temperatures of approximately 50 °C. Therefore, polycarbonate-diol-based self-healing PU containing a disulfide bond was synthesized in this study (Appendix A). To initiate self-healing triggered by the NIR laser, PU nanocomposites were prepared using PDA–rGO. Figure 2b shows the fabrication procedure of the PDA–rGO/PU nanocomposites containing different filler contents. GO was functionalized with PDA to enhance its compatibility with the polymer matrix and enhance the photothermal effect [23,39]. The thermal and chemical properties of the 1 wt% GO/PU and 0.1, 0.5 and 1 wt% PDA–rGO/PU nanocomposites were confirmed by DMA and FTIR spectroscopy.

Figure 5a shows the tan delta (δ) curves measured by DMA in the range of −100 °C to 150 °C. The δ peaks represent the structural transitions of the polymer. The glass transition temperature (*T*_g_) and melting transition temperature (*T*_m_) of PU were observed at −2.14 °C and 44.62 °C, respectively. The *T*_g_ of the PDA–rGO/PU nanocomposites increased as the filler content increased to 0.1, 0.5 and 1 wt%. In the case of the polyurethane nanocomposite, to which GO or PDA–rGO nanofiller was added, the Tg was similar or slightly increased, compared to the neat PU sample. This reason can be interpreted as the cause of the low content of the added nanofiller.

In addition, the *T*_g_ of the 1 wt % GO/PU nanocomposite was higher than that of the 1 wt% PDA–rGO/PU nanocomposites with the same filler content. This result indicates an interaction between the hard domain of neat PU and GO, showing that the thermal properties of GO/PU nanocomposite are substantially to those of neat PU, probably owing to the reinforcing effect of the homogeneous dispersion within the polymer matrix and the strong interaction between OH groups in GO and the main backbone of PU [40,41]. As can be seen in the FTIR spectra of Figure 5b, the characteristic peak of the hydrogen-bonded carbonyl group at 1703 cm^−1^ slightly increases in intensity.

The mechanical properties of the PU nanocomposites were evaluated using UTM equipment. Figure 6 shows the strain–stress curves of nanocomposites with different GO and PDA–rGO contents. Table 1 summarizes the thermal properties, breaking stress, elongation at break, and Young’s modulus for pure PU and PU nanocomposites. The pure PU film showed a 582.4% elongation at break and breaking stress of 18.1 MPa. Compared to pure PU, 1 wt% GO/PU nanocomposites showed a decreased elongation at break (433.8%) and breaking stress of 16.4 MPa. In contrast, the PDA–rGO/PU nanocomposite exhibited an increased Young’s modulus and breaking stress as the filler content increased, but the elongation at break decreased. In particular, the breaking stress and elongation at break of the 1 wt% PDA–rGO/PU nanocomposites were higher than those of the 1 wt% GO/PU nanocomposites with the same filler content. The PDA-coated GO was expected to be uniformly dispersed in the PU matrix [9,42], because the PDA-coated GO showed more stable dispersion stability in the solvent than GO (Appendix A). In addition, the FE–SEM image showing the cross-section of the nanocomposite revealed that the aggregated GO was dispersed in the polymer matrix, whereas the PDA–rGO was uniformly dispersed in the polymer matrix (in comparison to the 1 wt% GO/PU and PDA–rGO/PU nanocomposites) (Appendix A).

### 3.3. Photothermal and Self-Healing Properties of the PDA–rGO/PU Nanocomposites

Figure 7a shows the results of real-time measurement of the photothermal properties of GO and PDA–rGO in the polymer matrix when irradiated with a NIR laser with a laser power density of 0.1 W/cm^2^. Pure PU maintained a constant temperature of approximately 25 °C, regardless of whether the laser was on or off. In contrast, the temperature of the PDA–rGO/PU nanocomposites increased with increasing filler content. (PDA–rGO/PU nanocomposites with filler contents of 0.1, 0.5 and 1 wt% were measured at 55, 89 and 102 °C, respectively.) Polydopamine has been used as a photothermal agent with strong NIR absorption and high photothermal conversion efficiency [39,43]. Thus, the photothermal conversion increased as the PDA–rGO nanofiller content increased. To confirm the dopamine–functionalization effect, the photothermal properties of the GO and PDA–rGO/PU nanocomposites were observed at 1 wt% filler content. The maximum temperatures of the GO/PU and PDA–rGO/PU nanocomposites were 96 and 101 °C, respectively, indicating that the photothermal conversion efficiency of PDA–rGO was higher than that of GO, due to strong NIR absorption of polydopamine.

The self-healing efficiency was confirmed by scratch recovery of the PU nanocomposites. Damaged specimens were prepared according to a previous study [26]. The damaged nanocomposites were irradiated for 10 min with a NIR laser with a power density of 0.1 W/cm^2^. The strain–stress curves for the original and healed PU nanocomposites with 1 wt% GO and PDA–rGO are shown in Figure 7b. The 1 wt% GO/PU and 1 wt% PDA–rGO/PU nanocomposites showed healing efficiencies of 82.0% and 87.3%, respectively. In this study, we synthesized a self-healable polyurethane block copolymer consisting of a reversible covalent disulfide bond. In other words, if a temperature is applied when the polymer chain is cut due to damage, the disulfide bonds adjacent to each other form a radical intermediate, and then bond begins to heal. These stimuli have been directly supplied with energy until now, but in this study, the nanofillers inside the matrix were heated using near-infrared rays to convert light energy into thermal energy to enable healing. Thus, the photothermal conversion increased as the PDA–rGO nanofiller content increased. In particular, the PDA–rGO/PU nanocomposites showed a higher self-healing efficiency than the GO/PU nanocomposites, indicating that the nanofillers were well-dispersed in the polymer matrix by the functionalization of PDA. Therefore, heat transfer to the polymer matrix occurred uniformly, owing to the local surface plasmon resonance effect of PDA–rGO during NIR laser irradiation [9]. However, pure polyurethane did not heal, because it does not contain fillers that can generate heat after irradiation with NIR. For this reason, it is not included in Figure 7b.

## 4. Conclusions

To improve the compatibility with the polymer matrix as well as the photothermal effect, GO was reduced using PDA. XRD and Raman spectroscopy showed that PDA–rGO was successfully synthesized. PDA–rGO was mixed with the synthesized self-healing PU by the ex situ method. Photothermal conversion was investigated, based on the filler content. Compared with the same amount of GO, the introduction of PDA–rGO into the PU matrix improved the photothermal effect and mechanical properties. The self-healing efficiency of the damaged nanocomposite using the thermal energy generated by the NIR laser was 82.0% and 87.3% for GO and PDA–rGO, respectively. PDA–rGO/PU nanocomposites have enhanced self-healing ability, compared to GO/PU nanocomposites, owing to the homogenous dispersion between the filler and PU matrix and improved photothermal properties.

## Figures and Tables

**Figure 1 polymers-13-01274-f001:**
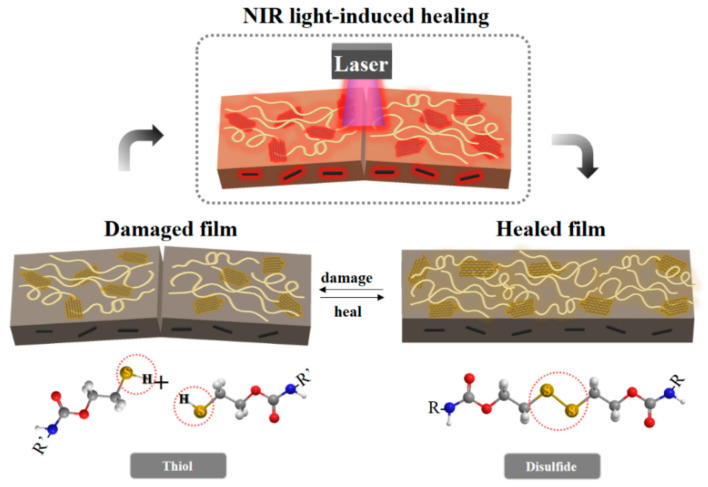
Schematic illustration of self-healing system using photothermal polydopamine-coated graphene oxide (PDA–rGO) triggered by near-infrared (NIR) laser.

**Figure 2 polymers-13-01274-f002:**
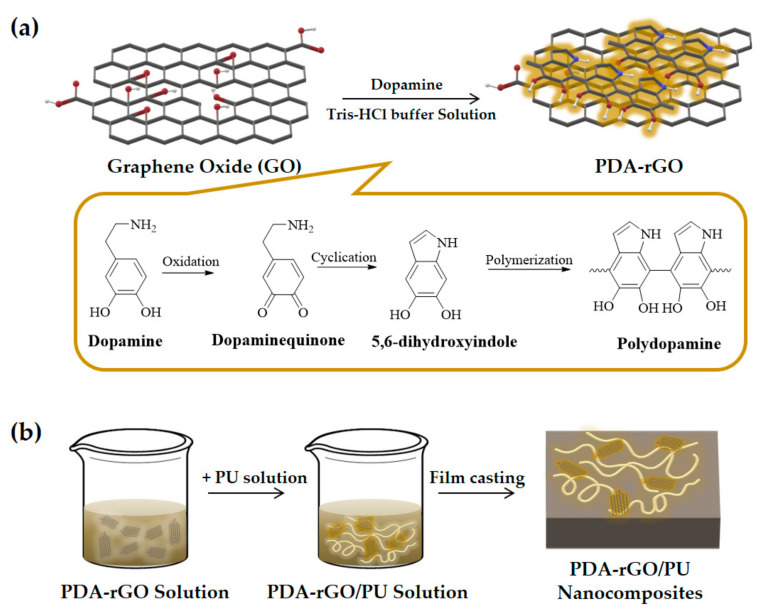
Schematic illustration of (**a**) the preparation of PDA–rGO, and (**b**) PDA–rGO/polyurethane (PU) nanocomposites.

**Figure 3 polymers-13-01274-f003:**
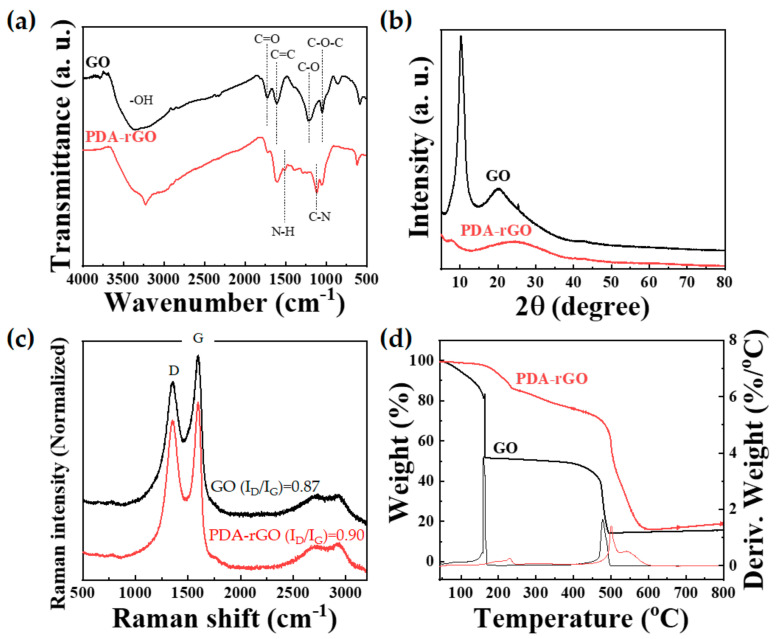
(**a**) FTIR spectra, (**b**) XRD patterns, (**c**) Raman spectra, and (**d**) TGA curves of graphene oxide (GO) and PDA–rGO.

**Figure 4 polymers-13-01274-f004:**
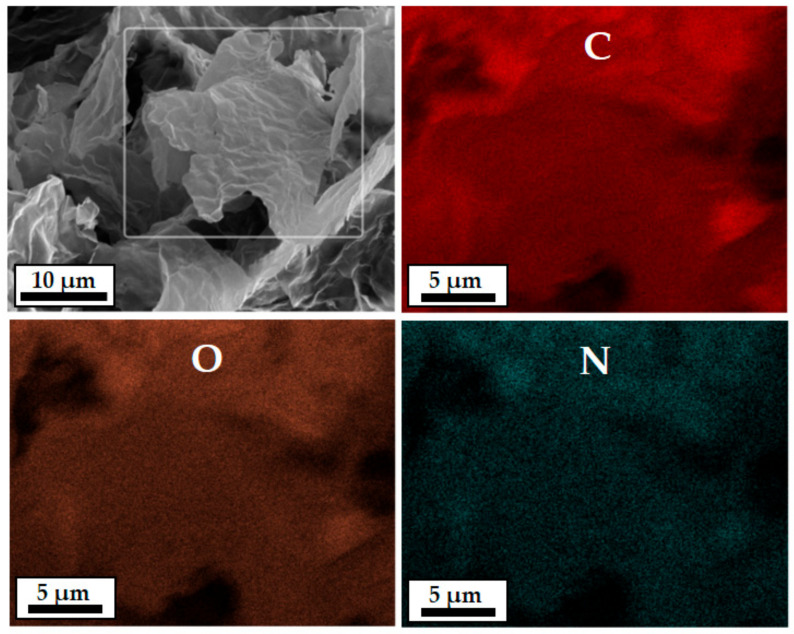
SEM–energy dispersive X-ray (EDX) mapping images showing the distribution of carbon (C), oxygen (O), and nitrogen (N) of PDA–rGO.

**Figure 5 polymers-13-01274-f005:**
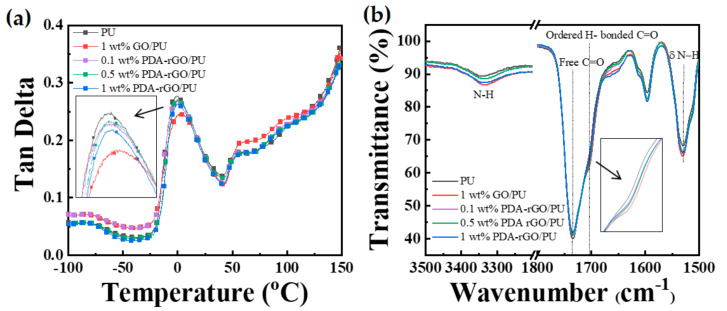
(**a**) Tan delta curves measured by dynamic mechanical analyzer (DMA) and (**b**) FTIR spectra of neat PU and its nanocomposites with 1 wt% of GO and 0.1, 0.5 and 1 wt% of PDA–rGO.

**Figure 6 polymers-13-01274-f006:**
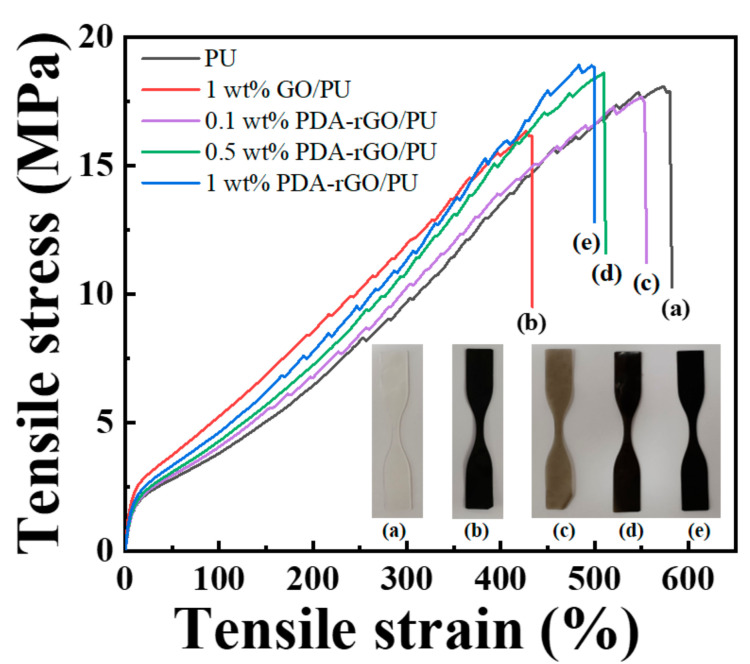
Strain–stress curves of (**a**) neat PU, (**b**) 1 wt% GO/PU, (**c**) 0.1 wt% PDA–rGO/PU, (**d**) 0.5 wt% PDA–rGO/PU, and (**e**) 1 wt% PDA–rGO/PU nanocomposites.

**Figure 7 polymers-13-01274-f007:**
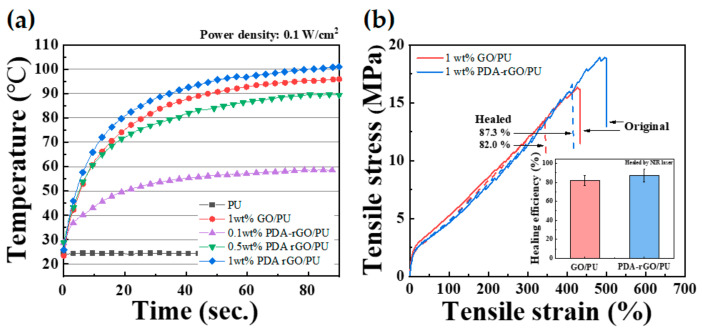
(**a**) Change in surface temperature during NIR light irradiation at a power density of 0.1 W/cm^2^ and (**b**) strain–stress curves for original and healed of 1 wt% GO/PU and 1 wt% PDA–rGO/PU nanocomposites. Self-healing efficiency inserted.

**Table 1 polymers-13-01274-t001:** Thermal and mechanical properties of PU and PU nanocomposites.

Sample Codes	T_g_ (°C)	T_m_ (°C)	Breaking Stress (MPa)	Elongation at Break (%)	Young’s Modulus (MPa)	Toughness (J/m^2^)
PU	−2.1	44.6	18.1 ± 0.65	582.4 ± 41.2	8.44 ± 0.86	5672.3 ± 129.1
1 wt% GO/PU	1.3	46.7	16.4 ± 0.87	433.8 ± 15.7	10.06 ± 2.17	3983.4 ± 99.8
0.1 wt% PDA–rGO/PU	−1.6	45.1	17.7 ± 0.28	555.7 ± 25.6	6.84 ± 0.29	5357.5± 107.3
0.5 wt% PDA–rGO/PU	−0.8	45.8	18.6 ± 0.86	511.7 ± 17.3	6.9 ± 1.67	4979.3 ± 101.6
1 wt% PDA–rGO/PU	−0.1	46.0	18.9 ± 0.24	500.0 ± 30.4	8.3 ± 1.17	5000.9 ± 113.9

## Data Availability

The data presented in this study are available on request from the corresponding author.

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
