# Peer review of "Rapid and Local Self-Healing Ability of Polyurethane Nanocomposites Using Photothermal Polydopamine-Coated Graphene Oxide Triggered by Near-Infrared Laser"

_polymers, 2021, doi:10.3390/polym13081274_

Round 1

Reviewer 1 Report

The manuscript entitled “Rapid and Local Self-healing Ability of Polyurethane Nanocomposites using Photothermal Polydopamine-Coated Graphene Oxide Triggered by Near-Infrared Laser” addresses the synthesis of a self-repairing polyurethane loaded with graphene oxide coated with a layer of polydopamine.

Despite the fact that the work is well written, the authors have made a light interpretation of the results, avoiding a justification between chemical structure and properties. Overall, the authors simply indicate whether the property values go up or down, without providing a deeper explanation to that result. Next I will indicate in more detail the weakest points of the work.

  • In Figure 3c, the authors show the Raman spectra of the pristine and polydopamine-coated graphene oxide samples. The authors have detected bands D and G and have measured the relationship between the two. The authors should have indicated the procedure for determining this relationship (Lorentzian deconvolution of the D and G bands?). As well as an deeper interpretation of the change in R value, due to the fact of having incorporated the polymer on the surface of GO.
  • In Figure 3d, the authors show the mass losses of pristine and coated GO. The authors mention that pristine GO has a first loss of 45% at 160ºC, due to the loss of oxygenated groups. However, in GO coated with polydopamine, the first loss is only 15%. Taking into account that the atmosphere used is air, it should be expected that the first loss would have been greater than 45%, due to the loss of the GO oxygenated groups and the combustion of the organic coating ... The authors should address this issue. Additionally, the authors should have discussed why such a significant change in GO thermal stability occurs.
  • Regarding the thermal transitions of polyurethane loaded with modified GO, only the authors indicate the temperatures at which changes are observed (Tg and Tm). However, a more appropriate interpretation of the results has been lacking, linking these transitions to the chemical composition of the soft or hard segment. Additionally, the authors indicate that the presence of GO charges produces an increase in Tg and a decrease in Tm, however, these increases are very slight and not very significant (pristine polymer -2.1ºC, charged polymer -0.1ºC). Therefore, this matter should also be approached accordingly and misinterpretation avoided.

On the other hand, the authors have previously studied this same type of thermal transitions for the same type of polyurethanes using DSC, instead of DMA (Macromolecular Research 2020, 28, 234-240.) In that case, the authors appreciated the thermal transitions associated with the soft and hard segment. These same types of interpretations should be transferred to this manuscript in order to increase its quality.

  • The authors indicate: "This is due to the increase in hydrogen bonding between the urethane groups of the polymer matrix and the abundant hydroxyl groups of GO". This phrase is referenced to a previous work of the authors, but in which the interactions of the hydroxyl groups of GO on a polymeric matrix were not precisely addressed. In addition, the polydopamine that covers the GO also has hydroxyl functions on the surface so it could also favor compibilization with the polymer. Therefore, the authors should find an appropriate reference for this fact or find another scientific justification that satisfies the experimental results.
  • Finally, the authors have evaluated the self-repair capacity of polyurethanes. Reviewing previous work by the authors, this reviewer found that these (unloaded) polyurethanes exhibited very high self-repair capabilities greater than 90%. The chemical structure of those polyurethanes was very similar to those presented in the manuscript. The absence of this comparison is not adequately addressed. In fact, a justifying interpretation comparing these results could provide a high scientific quality.

Finally, it should be noted that manucripts focused on self-repair should have a real reference system. In this case, if the authors indicate that the presence of dynamic disulfide bonds (-S-S-) are responsible for the self-healing capacity of the polymer, the synthesis of a polyurethane without this functional group must be included. In this way, it will be possible to demonstrate experimentally that disulfide bonds are responsible for self-repair. Lacking this reference, the capacity for self-repair could be partially linked to the macroscopic diffusion of the chains due to the heating of the sample by the NIR laser. Therefore, the disulfide bonds would not be responsible for self-repair.

Based on these points described above, I consider that the manuscript should not be accepted in the present form.

Author Response

Referee # 1

The manuscript entitled “Rapid and Local Self-healing Ability of Polyurethane Nanocomposites using Photothermal Polydopamine-Coated Graphene Oxide Triggered by Near-Infrared Laser” addresses the synthesis of a self-repairing polyurethane loaded with graphene oxide coated with a layer of polydopamine. Despite the fact that the work is well written, the authors have made a light interpretation of the results, avoiding a justification between chemical structure and properties. Overall, the authors simply indicate whether the property values go up or down, without providing a deeper explanation to that result. Next I will indicate in more detail the weakest points of the work.

[Comment 1] In Figure 3c, the authors show the Raman spectra of the pristine and polydopamine-coated graphene oxide samples. The authors have detected bands D and G and have measured the relationship between the two. The authors should have indicated the procedure for determining this relationship (Lorentzian deconvolution of the D and G bands?). As well as an deeper interpretation of the change in R value, due to the fact of having incorporated the polymer on the surface of GO..

Reply: Thanks to the referee’s appropriate comment. The D and G band are marked in Figure 3(c). Raman data was normalized based on the G band peak and R value was calculated as the intensity ratio of the D and G bands. PDArGo showed a higher R value than GO, and its interpretation was added as follows:

“The intensity ratio of the D and G bands (ID/IG), R value(defined as ID/IG, the integrated intensity of the D band divided by the integrated intensity of the G band), is associated with graphene disorder; therefore, it is used to measure the degree of defects. PDA-rGO showed a slight increase in D band intensity compared to GO. The R value of GO and PDA-rGO were 0.87 and 0.90, respectively. This results show that an increase in edge defects in PDA-rGO due to the grafting of PDA in GO.”

[Comment 2] In Figure 3d, the authors show the mass losses of pristine and coated GO. The authors mention that pristine GO has a first loss of 45% at 160ºC, due to the loss of oxygenated groups. However, in GO coated with polydopamine, the first loss is only 15%. Taking into account that the atmosphere used is air, it should be expected that the first loss would have been greater than 45%, due to the loss of the GO oxygenated groups and the combustion of the organic coating ... The authors should address this issue. Additionally, the authors should have discussed why such a significant change in GO thermal stability occurs.

Reply: Thanks to the referee’s critical comment. According to the referee’s appropriate comment, we have revised the thermal stabilities part of PDA-rGO:

“In contrast, PDA-rGO showed a slow weight loss of 15 wt% up to 240 °C, which represents the elimination of the organic functional group[30], and rapid weight loss of 58 wt% between 450 and 600 °C due to the elimination of the carboxylic group. These results indicate that PDA-rGO is more thermally stable than GO, and confirmed that GO was successfully reduced. The relatively slow decomposition from 200°C to 500°C can be considered as the effect of delaying the thermal decomposition by the PDA coating layer. In other words, as the tem-perature gradually increased to high temperature, the PDA coating layer formed char, which delayed heat transfer to the inside, resulting in a relatively low weight reduction rate.”

 [Comment 3] Regarding the thermal transitions of polyurethane loaded with modified GO, only the authors indicate the temperatures at which changes are observed (Tg and Tm). However, a more appropriate interpretation of the results has been lacking, linking these transitions to the chemical composition of the soft or hard segment. Additionally, the authors indicate that the presence of GO charges produces an increase in Tg and a decrease in Tm, however, these increases are very slight and not very significant (pristine polymer -2.1ºC, charged polymer -0.1ºC). Therefore, this matter should also be approached accordingly and misinterpretation avoided.

The authors indicate: "This is due to the increase in hydrogen bonding between the urethane groups of the polymer matrix and the abundant hydroxyl groups of GO". This phrase is referenced to a previous work of the authors, but in which the interactions of the hydroxyl groups of GO on a polymeric matrix were not precisely addressed. In addition, the polydopamine that covers the GO also has hydroxyl functions on the surface so it could also favor compibilization with the polymer. Therefore, the authors should find an appropriate reference for this fact or find another scientific justification that satisfies the experimental results.

Reply: We thank the reviewer for his/her helpful comments on our work. As suggested by this reviewer, we have carried out DMA analysis in order to understand the effect of GO and PDA-rGOs on the Tg of polyurethane (see Figure below). Then, we have revised the related sentence in the results and discussion section by including DMA data:

line 215 and 224: “In addition, the Tg of the 1 wt % GO/PU nanocomposite was higher than that of the 1 wt% PDA-rGO/PU nanocomposites with the same filler content. This is due to the increase in hydrogen bonding between the urethane groups of the polymer matrix and the abundant hydroxyl groups of GO [24].” changed “In the case of the polyurethane nanocomposite to which GO or PDA-rGO nanofiller were added, the Tg was similar or slightly increased compared to the neat PU sample. This reason can be interpreted as the cause of the low content of the added nanofiller.
In addition, the Tg of the 1 wt % GO/PU nanocomposite was higher than that of the 1 wt% PDA-rGO/PU nanocomposites with the same filler content. This result indicates a interaction between GO and the hard domain of neat PU, showing that the thermal properties of GO/PU nanocomposite are substantially superior to those of neat PU, probably owing to the reinforcing effect of the homogeneously dispersed within the polymer matrix and the strong interaction between OH groups in GO and the main backbone of PU [27].”

[Comment 4] Finally, the authors have evaluated the self-repair capacity of polyurethanes. Reviewing previous work by the authors, this reviewer found that these (unloaded) polyurethanes exhibited very high self-repair capabilities greater than 90%. The chemical structure of those polyurethanes was very similar to those presented in the manuscript. The absence of this comparison is not adequately addressed. In fact, a justifying interpretation comparing these results could provide a high scientific quality.

Finally, it should be noted that manucripts focused on self-repair should have a real reference system. In this case, if the authors indicate that the presence of dynamic disulfide bonds (-S-S-) are responsible for the self-healing capacity of the polymer, the synthesis of a polyurethane without this functional group must be included. In this way, it will be possible to demonstrate experimentally that disulfide bonds are responsible for self-repair. Lacking this reference, the capacity for self-repair could be partially linked to the macroscopic diffusion of the chains due to the heating of the sample by the NIR laser. Therefore, the disulfide bonds would not be responsible for self-repair.

Reply: Thanks to the referee’s critical comment. As suggested by this reviewer, we have revised the related sentence in the results and discussion section.

line 275 and 282: “This study synthesized a self-healable polyurethane block copolymer consisted of a reversible covalent disulfide bond. In other words, if a temperature is applied when the polymer chain is cut due to damage, the disulfide bonds adjacent to each other form a radical intermediate, and then bond again to heal. These stimuli have been directly supplied with energy until now, but in this study, the nanofillars inside the matrix are heated using near-infrared rays to convert light energy into thermal energy to enable healing. Thus, the photothermal conversion increased as the PDA-rGO nanofiller content increased.”

line 2787 and 289: “However, pure polyurethane did not heal because it does not contain fillers that can generate heat after irradiation with NIR. For this reason, it is not included in Figure 7(b).”

Reviewer 2 Report

The paper entitled "Rapid and Local Self-healing Ability of Polyurethane Nano-composites using Photothermal Polydopamine-Coated Graphene Oxide Triggered by Near-Infrared Laser" presents some results on the dynamic and healing performances of polymeric nanocomposites.

The introduction in quite general, I suggest to add specific comment on PU based healing systems and nanocomposites and provide a critical analysis of the current research on the topic. A couple of examples:

- Grande et al., Effect of the polymer structure on the viscoelastic and interfacial healing behaviour of poly(urea-urethane) networks containing aromatic disulphides

- Hernandez, et al., Effect of graphene content on the restoration of mechanical, electrical and thermal functionalities of a self-healing natural rubber

Furthermore, one of the interesting part is the Near-Infrared Laser healing approach. How this approach can be compared with traditional one? See “Bode et al., Characterization of self-healing polymers: From macroscopic healing tests to the molecular mechanism”.

Regarding the results and discussion section, a large portion is dedicated to the description of the material preparation (that is ok). On the other hand, the part on the characterization is quite poor of details, especially the one on the self-healing performances that should be the core of the research. This part needs to be enlarged and reviewed prior to publication.

Author Response

Referee # 2

The paper entitled "Rapid and Local Self-healing Ability of Polyurethane Nano-composites using Photothermal Polydopamine-Coated Graphene Oxide Triggered by Near-Infrared Laser" presents some results on the dynamic and healing performances of polymeric nanocomposites.

The introduction in quite general, I suggest to add specific comment on PU based healing systems and nanocomposites and provide a critical analysis of the current research on the topic. A couple of examples:

[Comment 1] Grande et al., Effect of the polymer structure on the viscoelastic and interfacial healing behaviour of poly(urea-urethane) networks containing aromatic disulphides. Hernandez, et al., Effect of graphene content on the restoration of mechanical, electrical and thermal functionalities of a self-healing natural rubber. Furthermore, one of the interesting part is the Near-Infrared Laser healing approach. How this approach can be compared with traditional one? See “Bode et al., Characterization of self-healing polymers: From macroscopic healing tests to the molecular mechanism”.

Reply: Thanks for your kind comment. As suggested by this reviewer, we have revised the related sentence in the introduction section.

[Comment 2] Regarding the results and discussion section, a large portion is dedicated to the description of the material preparation (that is ok). On the other hand, the part on the characterization is quite poor of details, especially the one on the self-healing performances that should be the core of the research. This part needs to be enlarged and reviewed prior to publication.

Reply: Thanks for your kind comment. According to the referee’s appropriate comment, we have revised the manuscript to explain. The revised part was highlighted in the MS.

Reviewer 3 Report

This study has performed adequate characterization from different aspects to the target materials. The acquired data is exhibited in a very organized manner and discussed properly. Only a small amount of improvements are necessary to be made before this manuscript can be published. Some further thoughts could be used for the design of experiments and eventually better supports toward the delivered conclusion. Please find some detailed questions, comments, suggestions correspondingly to specific part as follows.

Line 85: Please specify the molecular weight cut-off for the dialysis membrane.

Line 97: Please provide the yield for the colorless solid and other synthesized samples.

Line 98: Please describe the steps for the solvent casting method in detail for this study.

Line 103: How is the thickness of the resulting films measured? How good was the thickness control? If 300~400 indicate the range of the resulting thickness, have you considered other method, such as doctor blade, for better control?

Figure 4: It might be better to use more different color scheme for C and O.
Line 194~196: The difference in Tg cannot be readily observed just from Fig. 5(a). It is better to include the zoomed-in figure like Fig. 5(b) does.

Fig. 5(b): The zoomed-in figure could be further enlarged for more apparent observations for readers.

Table 1: The relative toughness for each material could also be estimated using the area below each stain-stress curve. It is also an important mechanical property when self-healing performances are investigated.

Line 247: These two experimental values of healing efficiencies for materials with and without PDA seem close to each other. What is the error bar for these numbers? For healing efficiency, have you considered to use other mechanical parameters to calculate this important value? What is the advantage to select tensile stress other than other mechanical parameters?

Author Response

Referee # 3

This study has performed adequate characterization from different aspects to the target materials. The acquired data is exhibited in a very organized manner and discussed properly. Only a small amount of improvements are necessary to be made before this manuscript can be published. Some further thoughts could be used for the design of experiments and eventually better supports toward the delivered conclusion. Please find some detailed questions, comments, suggestions correspondingly to specific part as follows.

[Comment 1] Line 85: Please specify the molecular weight cut-off for the dialysis membrane.

Reply: Thanks to the referee’s appropriate comment, we have added the information about dialysis tube in the manuscript. “Dialysis membrane (MWCO 6-8 kDa, Spectrum Laboratories, Inc., USA)”

[Comment 2] Line 97: Please provide the yield for the colorless solid and other synthesized samples.

Reply: Thank you for the referee’s kind comment. The yield of the dried polymer after precipitation in methanol is 95%. The polymer yield result was added to the experimental part. “After washing several times with methanol, it was dried at 30°C in vacuum, and the yield was 95%.”

[Comment 3] Line 98: Please describe the steps for the solvent casting method in detail for this study.

Reply: Thanks to the referee’s appropriate comment, we have revised the preparation of PDA-rGO/PU nanocomposites part, including detailed solvent casting method (amount of the polymer and solvent) as follows: “The resulting product was a colorless solid (1 g), which was re-dissolved in chloroform (9 mL). PU nanocomposites with PDA-rGO were prepared by the solvent casting method on glass dishes with 12 cm diameters. PDA-rGO/PU solutions were prepared with PDA-rGO at 0.1, 0.5, and 1 wt%. The calculated amount of PDA-rGO was dispersed in chloroform (20 mL) by horn-type sonication for 30 min, and then the 10 wt% PU solution was added to the PDA-rGO solution. The mixture was stirred for 24 h at room temperature and dried under vacuum at 50 °C for 48 h.”

[Comment 4] Line 103: How is the thickness of the resulting films measured? How good was the thickness control? If 300~400 indicate the range of the resulting thickness, have you considered other method, such as doctor blade, for better control?

Reply: Thank you for the referee’s kind comment. The thickness was measured with a Micrometer. The synthesized self-healable PU was difficult to detach after dried on a glass plate or dish. Therefore, we used a Teflon dish with a 12 cm diameter. The thickness of the film could be controlled by dissolving the same amount of polymer in a solvent.

[Comment 5] Figure 4: It might be better to use more different color scheme for C and O.

Reply: Thank you for the referee’s kind comment. We tried changing the EDX mapping image of Carbon and Oxygen, but the image was not clear. So we have not made additional modifications.

[Comment 6] Line 194~196: The difference in Tg cannot be readily observed just from Fig. 5(a). It is better to include the zoomed-in figure like Fig. 5(b) does. Fig. 5(b): The zoomed-in figure could be further enlarged for more apparent observations for readers.

Reply: Thanks to the referee’s appropriate comment, we have revised the zoomed-in Figure 5 (a) and (b).

[Comment 7] Table 1: The relative toughness for each material could also be estimated using the area below each stain-stress curve. It is also an important mechanical property when self-healing performances are investigated.

Reply: Thanks to the referee’s appropriate comment, we have added the toughness of sample in Table 1.

[Comment 8] These two experimental values of healing efficiencies for materials with and without PDA seem close to each other. What is the error bar for these numbers? For healing efficiency, have you considered to use other mechanical parameters to calculate this important value? What is the advantage to select tensile stress other than other mechanical parameters?

Reply: Tensile strength tests of the origin film and the healed film were measured four times per sample. Error bars represent deviation values of the calculated self-healing efficiency, respectively. Tensile stress is an important factor in the physical properties of polymers and was used to confirm the self-healing efficiency in this study.

Round 2

Reviewer 1 Report

Based on the answers provided by the authors, I still consider that the manuscript should be rejected. The manuscript still has some inconsistencies that should be addressed correctly. Here is the justification for my conclusion.

In an oxidizing atmosphere (such as air), organic polymeric coatings, such as polydopamine, will show the highest mass loss before 350°C. In fact, in manuscript “Polydopamine-wrapped carbon nanotubes to improve the corrosion barrier of polyurethane coating”, (RSC Adv., 2018, 8, 23727, DOI: 10.1039/c8ra03267j) in figure 5, it can be seen how the organic coating of Polydopamine loses 60% by weight at 350ºC. Additionally, the same figure shows how the hybrid material (nanotubes + PDA) shows an intermediate mass loss between the pristine nanotubes and the PDA coating. These results would be as expected. Why have not these authors appreciated the protective effect of the PDA coating? The proposed justification "the PDA coating layer formed char, which delayed heat transfer to the inside, resulting in a relatively low weight reduction rate" still lacks previous works to support it and also lacks physical sense. Indeed, if the authors were right, they should also incorporate the TGA of the PDA coating alone and appreciate that effect they mention ... where one should expect that the increased stability should also manifest itself.

Reference 27 proposed by the authors continues to be their previous work and lacks the necessary information to conclude the following: “the reinforcing effect of the homogeneously dispersed within the polymer matrix and the strong interaction between OH groups in GO and the main backbone of PU” because GO has not been used in that paper. Additionally, the justification still does not make sense because OH groups are also present in the PDA organic coating. Therefore, the answer is not justified.

It is evident that the unloaded polyurethane was not capable of self-healing when irradiated with NIR light due to the absence of fillers. However, this same polyurethane could be self-healing if heat were used, in that case, the self-repair capacity could be comparable to those shown in this manuscript. My theory is based on previous work done by the authors (DOI 10.1007/s13233-020-8027-y). It would be very interesting to address this issue by comparing both approaches (thermal self-repair and NIR-induced self-repair).

On the other hand, the capacity for self-repair based on disulfides is well known, but my comment was focused on the synthesis of a polyurethane lacking these disulfides but loaded with fillers. For the synthesis of this new reference polyurethane, the authors should substitute disulfide-monomer for instance 1,4-butanediol. In this way, the authors could confirm that self-repair is due to a combined effect of disulfide bridges and polydopamine-coated fillers. In articles focused on self-repair, there is a general lack of appropriate reference systems. This issue must be addressed to demonstrate unequivocally that polymers self-repair due to the conclusions proposed by the authors. Therefore, the answer proposed by the authors is still not justified.

Reviewer 2 Report

Revisited paper improved the previous version.
